# A Deep Learning Framework for Real-Time Bird Detection and Its Implications for Reducing Bird Strike Incidents

**DOI:** 10.3390/s24175455

**Published:** 2024-08-23

**Authors:** Najiba Said Hamed Alzadjail, Sundaravadivazhagan Balasubaramainan, Charles Savarimuthu, Emanuel O. Rances

**Affiliations:** College of Computing and Information Sciences, University of Technology and Applied Sciences-AL Mussanah, Muladdah P.O. Box 191, Oman; charles@act.edu.om (C.S.); emanuel@act.edu.om (E.O.R.)

**Keywords:** deep learning, bird strike, CNN, UAV, R-FCN, YOLO

## Abstract

Bird strikes are a substantial aviation safety issue that can result in serious harm to aircraft components and even passenger deaths. In response to this increased tendency, the implementation of new and more efficient detection and prevention technologies becomes urgent. The paper presents a novel deep learning model which is developed to detect and alleviate bird strike issues in airport conditions boosting aircraft safety. Based on an extensive database of bird images having different species and flight patterns, the research adopts sophisticated image augmentation techniques which generate multiple scenarios of aircraft operation ensuring that the model is robust under different conditions. The methodology evolved around the building of a spatiotemporal convolutional neural network which employs spatial attention structures together with dynamic temporal processing to precisely recognize flying birds. One of the most important features of this research is the architecture of its dual-focus model which consists of two components, the attention-based temporal analysis network and the convolutional neural network with spatial awareness. The model’s architecture can identify specific features nested in a crowded and shifting backdrop, thereby lowering false positives and improving detection accuracy. The mechanisms of attention of this model itself enhance the model’s focus by identifying vital features of bird flight patterns that are crucial. The results are that the proposed model achieves better performance in terms of accuracy and real time responses than the existing bird detection systems. The ablation study demonstrates the indispensable roles of each component, confirming their synergistic effect on improving detection performance. The research substantiates the model’s applicability as a part of airport bird strike surveillance system, providing an alternative to the prevention strategy. This work benefits from the unique deep learning feature application, which leads to a large-scale and reliable tool for dealing with the bird strike problem.

## 1. Introduction

Aviation safety and wildlife, including birds, has been a subject of concern and research for many years within the aviation industry [1,2]. Bird strikes, which are a relatively common phenomenon, a serious threat to aviation security, and an expensive problem, have been increasing with the increase in air traffic [3]. This article presents the improvement in the deep learning model that will be used to sense and avoid bird strikes in airports’ surrounding areas.

### 1.1. Background

Bird strikes have been an unsolvable problem of aviation since its beginning: the first moment it occurred was in 1905 [4]. Such accidents can cause irreversible destruction of an aircraft, operational disturbances that are considered moderate, and even the chance of a catastrophic failure, posing a threat to human lives [5,6,7]. Pursuant to the Federal Aviation Administration (FAA), bird strikes have been the major cause of damage to US civil and military transportation service estimated in the annual amount of USD 600 million in 2010, and now the costs will be higher [8]. The increase in global air traffic and variations in the population of birds due to environmental changes are other reasons for avian flu spread [9].

The array of solutions used in bird strike avoidance ranges from habitat management to avian radar systems and acoustic deterrents [10]. Nevertheless, they appear to be inadequate in terms of sensing and flexibility when different environmental conditions are put in place. This explains why there is an urgent need to look at better and superior alternatives.

The authors [11] aimed to develop a prediction model using machine learning techniques such as fuzzy decision trees and dynamic neural networks, which are quite useful in the aviation safety system and were adopted to give a prognosis for the failures in the aircraft beforehand to increase aviation safety systems and decrease aircraft crash rates. The results of the prediction are within a range of 80% to 90%. The synthetic dataset testing against the proposed aircraft crash prediction model was one of the ways to test the proposed aircraft crash prediction model.

In another research study [12], a new highly challenging dataset, Air Birds, has been presented, formed by 118,312 images obtained by an aerial camera. It has a complete set of 409967 GPS-like tags attached manually to each flying bird captured in the image. The average dimension of all the annotated objects has 10 pixels as the smallest one, following 1920 × 1080 images. The dataset consists of pictures of the airport obtained by four cameras installed in a network in four seasons of 1 year. It features both birds on the ground and in flight, changes in lighting, and 13 meteorological situations. In the work of [13], the researchers presented a sample of drone and bird images in various environments and habitats. The dataset was curated using the Rob flow software which allowed us to custom edit the images using AI-assisted bounding boxes, polygons, and instance segmentation. A variety of input and output formats, including import and export, were supported by the software that could provide an interface with different machine learning frameworks. To attain the best accuracy, each image was manually segmented from edge to edge and detailed for training the YOLO-based model.

This research in [14] fully utilized a deep learning-based object detection structure by aerial photographs with an Unmanned Aerial Vehicle (UAV). The aerial photography dataset used included different birds’ photos and those taken in various habitats, near lakes, and adjacent to farmland. Moreover, the aerial view of the placements of the bird decoys was filmed to obtain more birds’ information. Bird detection or classifying models using Faster R-CNN, R-FCN, SSD, Retina net, and YOLO were created, and their time complexity as well as precision were evaluated by making comparisons between them. The test results revealed that the Faster R-CNN performs better than YOLO in terms of accuracy while YOLO is faster among the models.

The authors of [15] developed a unified spatiotemporal CNN model for visual object segmentation which was composed of the spatial segmentation and temporal coherence networks seeing as they achieved accurate results on challenging datasets [16,17,18,19]. The authors of [20] tested feature extraction variation and spatiotemporal convolution deep neural network (STCNN) for driver fatigue detection. The combination of the two produced a better result than either feature extraction or STCNN both used independently, as revealed by the experiments. The attention, time, and convolutional neural network sequence that was inspired by an attention mechanism of CNN is a component of the proposed STCNN architecture. In another study [21], a new approach to the aviation failure problem was suggested that comprised STL (seasonal trend decomposition losses) and a hybrid model that uses a transformer and ARIMA components. The combination of STL and ARIMA was a good idea to predict aviation failure.

Previous approaches have largely relied on simpler CNNs or traditional machine learning algorithms that do not account for the temporal dynamics of moving objects. Such models lack the sophistication needed to differentiate between birds and other moving objects like aircraft, leading to significant detection inaccuracies. Additionally, these systems often require extensive manual tuning and are not scalable to different or larger datasets without substantial reconfiguration.

The fundamental research for aviation safety gravitates around machine learning models for aircraft fault estimation and object detection; however, there are insufficient studies particularly on real-time bird detection to prevent bird strike incidents. The present study has taken this gap to its fill by introducing new feature extraction methods coupled with deep learning techniques.

### 1.2. Motivation

Deep learning as a revolutionized AI technique has demonstrated outstanding abilities of breaking down and looking through visual data using CNNs and other such methods, making it apt for bird discovery in the densely crowded areas/airport locality. The inspiration in the neural network powerfulness in the field comes from this experiment and its objectives are to apply deep learning to the real-time identification of birds, which is vital for safety equipment in airplanes. The motivation of this research work is rooted from the need to improve the existing bird detection systems, which are often limited by their detection range and accuracy.

### 1.3. Problem Definition

The identification of precise, timely bird detection in and around airport areas with the aim of preventing bird strikes thus becomes the focus of this study. The complexity is in the multitude of bird species with different sizes, flight patterns, and behaviors, as well as in the unusual airport environment with moving aircraft, cars, and various types of infrastructure. Determining a model capable of classifying birds from fast-changing backgrounds under different weather conditions and times of day is a great difficulty.

### 1.4. Objectives

The main objective of the paper is to advance bird strike prevention technologies through the development of a novel deep learning framework optimized for real-time bird detection in aviation environments. The key goals supporting this objective are as follows:To develop spatiotemporal convolutional neural network (ST-CNN) mechanisms for enhanced feature recognition and dynamic temporal processing for accurate motion tracking.To optimize model performance metrics by fine-tuning the model’s parameters, ensuring minimal false positives and negatives for reliable detection.

### 1.5. Novel Approach

The technique suggested in this research is a novel ST-CNN which incorporated both spatial and temporal characteristics to identify birds. Using ideas originating from the analysis of EEG signals [20] for the detection of driving fatigue, the model introduces spatial attention with temporal dynamics. This makes it easier to detect the movement patterns of birds and minimize false measurements. The proposed technique makes the model capable of discarding noise information while targeting specific information relative to birds in flight such as flapping, and direction changes of wings are picked from the video feed.

### 1.6. Structure of the Paper

The remaining section of this paper is organized as follows: Section 2 describes the development of the model and the experimental methodology. Section 3 presents the findings of the performance evaluation of the proposed model and the comparisons with other existing models. Finally, Section 4 concludes with the highlight of the research contribution and the possible future research on the proposed model.

## 2. Methodology and Methods

Towards having a perfect solution to the bird strike problem, proper consideration was taken in the following aspects: proper collection of the dataset and creation of a DL model. This section defines and describes further the basic components of the proposed study, ranging from data collection methods down to the modeling method. The flow of the proposed research is illustrated in Figure 1.

### 2.1. Data Collection and Preprocessing

Given that deep learning models rely on the quality and variety of training data, then “the more data, the better the model”. To this end, a large sample of bird images belonging to various species, flying poses, and parts of the environment was collected. To supplement the dataset’s variety, additional techniques of image scaling, rotation, and altering the lighting were used to simulate different weather, time, and the bird’s distance from the camera. Table 1 discusses the parameters used in data preprocessing.

### 2.2. Model Architecture Design

To resolve the limitations of the current models, the proposed model is a spatiotemporal convolutional neural network that integrates spatial attention mechanisms and temporal analysis into one system. The model architecture is based on the latest works on using EEG signals to detect driver fatigue, generalized for the visual input [20]. The model consists of two main components: a spatially aware convolutional network (SACN) as well as an attention-based temporal analysis network, named ATAN.

#### 2.2.1. Attention-Based Temporal Analysis Network (ATAN)

ATAN is intended to work with the reiterated image data, while temporal features of birds’ motion describe the flow of data. It uses forward–backward LSTM layers, further improved with attention activities, to pay attention to essential temporal features reflecting the flight of birds like the flapping of wings and fluctuations in speed. The overall process of the proposed ATAN is shown in the following Figure 2.

Let $S=\{I_{1},I_{2},...,I_{T}\}$ denote a sequence of image frames captured over time T. Here, each frame $I_{t}$ represents a snapshot at time t. These frames are processed individually to extract spatial features before being analyzed temporally. For each frame $I_{t}$, we define the extracted spatial features as $F_{t}$. These are obtained via a convolutional neural network (CNN), which processes each frame as follows:(1)$$F_{t}=CNN\left(I_{t}\right)$$

The sequence of feature maps $\left\{F_{1},F_{2},...,F_{T}\right\}$ is then fed into a bidirectional LSTM network to capture the temporal dynamics of the birds in flight. The bidirectional LSTM allows the network to have both forward-looking and backward-looking insights at each time step, capturing patterns that unfold over time. Let $H_{t}^{fwd}$ and $H_{t}^{bwd}$ denote the hidden states of the forward and backward LSTMs at time t, respectively:(2)$$H_{t}^{fwd}={LSTM}^{fwd}\left(F_{t},H_{t-1}^{fwd}\right)$$
(3)$$H_{t}^{bwd}={LSTM}^{bwd}\left(F_{t},H_{t+1}^{bwd}\right)$$

The combined hidden state at each time t is then given by concatenating the forward and backward hidden states. Equation (4) is obtained by combining Equations (2) and (3).
(4)$$H_{t}=\left[H_{t}^{fwd};H_{t}^{bwd}\right]$$

With the temporal features extracted, an attention mechanism is applied to weigh the significance of each time step’s features based on their relevance to the detection task. The attention weights $\alpha_{t}$ are computed using a softmax function over the energy scores $e_{t}$, which are derived from the hidden states:(5)$$e_{t}=v_{a}^{T}tanh\left(W_{a}H_{t}+b_{a}\right)$$
(6)$$\alpha_{t}=\frac{exp\left(e_{t}\right)}{\sum exp\left(e_{k}\right)}$$
where $v_{a}$, $W_{a}$, and $b_{a}$ are trainable parameters of the attention network. The context vector C, which represents a weighted sum of the hidden states, is then calculated:(7)$$C=\sum \alpha_{t}H_{t}$$

This context vector C is a dynamic representation of the entire sequence, emphasizing the most relevant temporal features for bird detection. Finally, the context vector C is passed through a fully connected layer for classification. The output layer provides a probability distribution over two classes, “bird” and “no bird”:(8)$$P=softmax\left(W_{c}C+b_{c}\right)$$
where $W_{c}$ and $b_{c}$ represent the weights and bias of the output layer, and P provides the probabilities of bird presence in the sequence of frames.

#### 2.2.2. Spatially Aware Convolutional Network (SACN)

SACN focuses on analyzing individual frames for spatial features, utilizing a series of convolutional layers followed by spatial attention mechanisms. This component is tasked with identifying bird-specific features against complex airport backgrounds, effectively distinguishing birds from other moving objects like aircraft and vehicles. Figure 3 represents the workflow of the SACN mechanism.

Let $I=\{I_{1},I_{2},...,I_{N}\}$ represent a batch of N image frames where each frame $I_{n}$ corresponds to a different point in time or a different camera view, with n∈{1,2,...,N}. Each frame $I_{n}$ undergoes a series of convolutional operations that extract features at multiple levels of abstraction. We define the output feature map after the $l^{th}$ convolutional layer for frame $I_{n}$ as $F_{n}^{l}$. The convolutional operation can be mathematically represented as follows:(9)$$F_{n}^{l}=\sigma \left(W^{l}*F_{n}^{l-1}+b^{l}\right)$$
where “∗” denotes the convolution operation, $W^{l}$ and $b^{l}$ are the weights and biases of the $l^{th}$ convolutional layer, $F_{n}^{l-1}$ is the input feature map from the previous layer with $F_{n}^{0}=I_{n}$, and σ is the non-linear activation function, such as ReLU.

After each convolutional layer, a pooling layer is typically applied to reduce spatial dimensions and enhance the invariance of the features. The pooling operation for layer l is denoted as $P_{n}^{l}$ and is defined as follows:(10)$$P_{n}^{l}=pool\left(F_{n}^{l}\right)$$
where pool represents the pooling function, and here, max pooling is used. This is the most used type of pooling in CNNs. It reduces the dimensionality of feature maps by applying a maximum filter to non-overlapping subregions of the initial feature map. For each such subregion, max pooling takes the maximum value:(11)$$P_{n}^{l}\left(i,j\right)=\underset{\left(k,m\right)\in W\left(i,j\right)}{\max}F_{n}^{l}\left(k,m\right)$$
where $W\left(i,j\right)$ is the window of the feature map $F_{n}^{l}$ covered by the pooling operation at position $\left(i,j\right)$.

To emphasize relevant features and suppress less useful information, a spatial attention mechanism is applied to the feature maps. The mechanism generates a spatial attention map $A_{n}^{l}$ which is element-wise multiplied with the feature map to weight the importance of each feature spatially:(12)$$A_{n}^{l}=\sigma \left(Conv\left(F_{n}^{l},W_{a}^{l}\right)\right)$$
(13)$$F_{n}^{l,att}=A_{n}^{l}⊙F_{n}^{l}$$
where Conv is a convolution operation specific to the attention mechanism, $W_{a}^{l}$ are the weights of the attention convolution, and ⊙ represents element-wise multiplication.

The feature maps from different layers can be integrated using a feature integration function ϕ, which combines information across layers to form a rich representation of the spatial features:(14)$$F_{n}^{integrated}=\phi \left(F_{n}^{1,att},F_{n}^{2,att},...,F_{n}^{L,att}\right)$$

The integrated feature map $F_{n}^{integrated}$ is then flattened and passed through one or more fully connected layers for classification. The output is a probability score indicating the likelihood of a bird’s presence:(15)$$P_{n}=softmax\left(W_{c}F_{n}^{integrated}+b_{c}\right)$$
where $W_{c}$ and $b_{c}$ are the weights and bias of the fully connected layer.

The SACN is trained using a supervised learning approach where the loss function measures the discrepancy between the predicted probabilities and the ground truth labels. The loss function $L_{SACN}$ is typically a cross-entropy loss defined as follows:(16)$$L_{SACN}=-\sum y_{n}log\left(P_{n}\right)+\left(1-y_{n}\right)log\left(1-P_{n}\right)$$
where $y_{n}$ represents the true label for frame $I_{n}$, with $y_{n}=1$ indicating the presence of a bird and $y_{n}=0$ otherwise.

#### 2.2.3. Integration and Fusion

The outputs of ATAN and SACN are fused using a custom integration layer, combining temporal and spatial insights to make a final prediction. This fusion approach ensures that the model’s decision-making process benefits from a comprehensive understanding of both the spatial appearance and temporal behavior of birds in flight. Assume that the ATAN and SACN output feature maps for the $n^{th}$ frame are represented as $C_{n}$ (the context vector from ATAN) and $F_{n}^{integrated}$ (the integrated feature map from SACN), respectively. The work flow of ATAN and SACN is given in the Figure 4. 

The goal of the fusion layer is to combine these two sets of features in a way that preserves and utilizes the information from both sources. This is conducted through a fusion function ψ, which is defined as follows:(17)$$F_{n}^{fused}=\psi \left(C_{n},F_{n}^{integrated}\right)$$

A common choice for ψ is concatenation followed by a fully connected layer that mixes the features, although other fusion techniques like summation or multiplication could also be considered. If concatenation is used, the fusion function can be mathematically expressed as follows:(18)$$F_{n}^{fused}=W_{fused}\cdot \left[C_{n};F_{n}^{integrated}\right]+b_{fused}$$
where $W_{fused}$ and $b_{fused}$ are trainable weights and biases of the fusion layer, and [;] denotes the concatenation operation.

The fused feature vector $F_{n}^{fused}$ is then fed into a classification layer to make a final prediction about the presence of a bird in the frame. The output of the classification layer is a probability distribution over possible classes, given by the following:(19)$$P_{n}=softmax\left(W_{class}F_{n}^{fused}+b_{class}\right)$$
where $W_{class}$ and $b_{class}$ are the weights and bias of the classification layer, and $P_{n}$ represents the predicted probabilities.

During training, a loss function is employed to optimize the model parameters. Given that the model is now performing a fusion of temporal and spatial features, the loss function $L_{total}$ is a combination of losses from both ATAN and SACN as well as a classification loss from the fused output:(20)$$L_{total}=\lambda_{1}L_{ATAN}+\lambda_{2}L_{SACN}+\lambda_{3}L_{class}\left(P_{n},y_{n}\right)$$
where the coefficients $\lambda_{1}$, $\lambda_{2}$, and $\lambda_{3}$ are hyperparameters that balance the contributions of the respective components. The classification loss $L_{class}$ for the fused feature vector is typically a cross-entropy loss if the task is a classification problem, defined as follows:(21)$$L_{class}=-\sum \left(y_{n}log\left(P_{n}\right)+\left(1-y_{n}\right)log\left(1-P_{n}\right)\right)$$
where $y_{n}$ is the true label for the presence or absence of a bird in the $n^{th}$ frame.

### 2.3. Proposed Algorithm

In the framework of this current study, we focus on preventing bird strikes through real-time detection. The proposed work develops a comprehensive algorithmic approach that seamlessly integrates with the spatiotemporal convolutional neural network architecture detailed in Section 2.2. This section describes the procedural aspects of the proposed approach. It encompasses image preprocessing, bird detection, and the fusion of spatial and temporal data for enhanced decision-making.

#### 2.3.1. Image Preprocessing and Augmentation

The initial stage of the proposed algorithm involves meticulous preprocessing and augmentation of raw image data. Given a batch of raw images I, each image Ii is first resized to a uniform dimension, ensuring consistency across the dataset. Following that, each image undergoes normalization to standardize pixel values, by preparing the data for efficient model processing. Augmentation plays a crucial role in enhancing the robustness of the model. Through techniques such as random flipping, rotation by an angle θ, and scaling by a factor s, it simulates a variety of environmental conditions. This diversity in the training data is instrumental in developing a model capable of accurate bird detection across different scenarios. The process of data preprocessing is given in the Algorithm 1.
**Algorithm 1:** Image Preprocessing and AugmentationVariables:
 -Let I be a batch of raw images $I=\{I_{1},I_{2},...,I_{n}\}$. -${I^{’}}_{i}$ represents the $i^{th}$ preprocessed image. -θ is the rotation angle. -s is the scaling factor for resizing. -p is the probability of applying a particular augmentation.Algorithm: Preprocess_and_Augment_Images(I)
 1.Initialize an empty list $I^{’}=\{\}$ for preprocessed images. 2.For each image $I_{i}$ in I:
   a.Resize $I_{i}$ to a fixed dimension, yielding $I_{i,resize}$.   b.Normalize $I_{i,resize}$ to obtain $I_{i,norm}$.   c.For each augmentation operation *op* (e.g., flip, rotate):Generate a random number r between 0 and 1.If $r<p_{op}$, apply *op* to $I_{i,norm}$, resulting in $I_{i,op}$.   d.Add $I_{i,op}$ (or $I_{i,norm}$ if no *op* was applied) to I′. 3.Return I′.

#### 2.3.2. Bird Detection Model Prediction

After preprocessing, the images are inputted to the deep learning model “M” to detect a bird. The model’s prediction result includes scores “P” for every possible bird detection in an image and the corresponding bounding boxes “B”. If a detection’s confidence is above or equal to the predetermined T, then it is considered valid and unique and therefore used in the subsequent investigation. Therefore, to improve the detection outcomes and remove duplicate detections, the method uses Non-Maximum Suppression (NMS). This step is crucial in arriving at the most likely detections of birds while at the same time minimizing the number of false positives of the model. Furthermore, NMS is pivotal in enhancing the accuracy and reliability of the bird strike prevention system. It minimizes false positives and overlaps in detected birds, providing clearer and more actionable insights for aviation safety measures.

A distinctive feature of the proposed approach is the integration and fusion of temporal and spatial data. It is derived from ATAN and SACN, respectively. This fusion is facilitated by a custom integration layer which combines the context vector C from ATAN with the integrated feature map F integrated from SACN.

The fusion function ψ amalgamates these data sources, leveraging both the spatial and temporal characteristics of bird movements. This comprehensive data integration ensures that our model’s predictions are based on a holistic understanding of birds in flight, significantly improving detection accuracy. The process of bird detection model is given in Algorithm 2. Further, Algorithm 3 presents the step-by-step process of finding the non-maximum suppression of the calculated data.
**Algorithm 2:** Bird Detection Model PredictionVariables:- D represents the set of detection results.- P is the set of prediction scores.- B is the set of bounding boxes.- T is the detection threshold.- Iprocessed is a preprocessed image.- M represents the deep learning model.Algorithm: Detect_Birds(Iprocessed,M,T)1. Pass Iprocessed through model M to obtain predictions P and bounding boxes B.
2. Initialize an empty set for detection results D={}.3. For each pi, bi in P, B:   a. If pi > T:     i. Add (bi,pi) to D.4. Apply Non-Maximum Suppression (NMS) to D, yielding Dfiltered.5. Return Dfiltered.

**Algorithm 3:** Non-Maximum Suppression (NMS)Variables:
 -Let D={d1,d2,...,dn} represent the set of detected bounding boxes, where each detection di is associated with a confidence score si and a bounding box bi. -Dfinal is the final set of detections after applying NMS. -IoUbi,bj represents the Intersection over Union between the bounding boxes bi and bj. -TIoU is the IoU threshold for determining overlap.
1. Sort Detections: Sort all detections D in descending order based on their confidence scores si.  Dsorted=sortD,key=si,order=descend2. Initialize Final Detections: Set Dfinal={}.3. Iterate Over Sorted Detections:  - For each detection di in Dsorted:   - Compare di with each detection dj in Dfinal to calculate the IoU:    IoUbi,bj=areabi∩bjareabi∪bj   - If IoUbi,bj>TIoU for any j, di is considered redundant and is not added to Dfinal.4. Add Non-Redundant Detections to Final Set:  - If dj is not redundant, add it to Dfinal:    Dfinal=Dfinal∪{di}5. Return Dfinal as the set of detections after applying NMS.

### 2.4. Training and Validation

The data were augmented to increase the size of the data and the data were split in the ratio of 7:3 with 70% of the augmented data used for training the model and the remaining 30% used for the validation. Training involved employments of advanced optimization algorithms and loss functions in the contrary object detection tasks. A validation process is concerned with accuracy, speed, reliability, and performance under different environmental situations, whereby the metrics applied include precision, recall, and the F1 score.

### 2.5. Dataset and Experimental Setup

For the study on bird strike prevention technology, a valuable source for the research work was the construction of a proper database meant for the differentiation of the birds from the UAVs [22]. This images file contains 20,925 labelled images with dimensions of 640 × 640, with birds and drones taken in various settings and lighting. All the images are properly divided and labeled following the YOLOv7 specifications to facilitate the accurate identification of the objects during the training of the deep learning models. With sets for training and validation as well as for testing the model performance, the dataset provides comprehensive and optimized assistance in training and fine-tuning the model specifically for real-life applications, and every image in the dataset is thoroughly annotated, allowing for the precise detection of airborne objects in real-time conditions. Table 2 presents the hardware and software used for the experiments. The deep learning parameters are presented in Table 2.

## 3. Results

Table 3 below identifies a comparison of the results of performance measurements between the developed model and six benchmark meta-architectures, namely, Faster R-CNN, R-FCN, Retina Net, SSD, YOLO v4, and YOLO v5. The assessment measures used are precision, recall, F1 score, accuracy, the area under the curve of the receiver operating characteristic (AUC-ROC), and the intersection over union (IoU). The following work shows a higher efficiency in all investigated characteristics, which speaks for its possible effectiveness in bird detection in the sphere of aviation security. It is notable that the following model developed from the proposed model enjoys percentages of precision of 0. Although, we obtain a satisfactory precision of 97% and an accuracy of 96%, thereby re-asserting the tool’s capability to make accurate threat assessments on birds.

A more detailed understanding of how the proposed model performs in terms of real-time bird detection is provided with the help of the evaluation of the inference time and the confusion matrix which can be found in Figure 5 and Figure 6, respectively.

In more detail, Figure 5 shows the comparative analysis of the proposed model with other competitive meta-architectures such as Faster R-CNN, R-FCN, Retina Net, SSD, YOLO v4, and YOLO v5 in terms of inference time. The inference time, which is in milliseconds (ms), can be explained by how quickly the system needs to decide in real-time applications ranging to the extent of preventing bird strikes in aviation. A competitive inference time of roughly 45 ms is demonstrated, thus enhancing the viability of the model for real-time analysis. Compared to other described architectures, rapid processing capability means that the model is optimized for speed without a significant loss of the detection accuracy, which is quite suitable for the usage in the aircraft safety systems where the immediate response matters.

The model performance of the proposed work has been quantified by using a confusion matrix and this is depicted in Figure 6. The confusion matrix shows the number of true positives, true negatives, false positives, false negatives, and the number of objects that are recognized as a bird and the number of objects that are not recognized as bird objects, meaning it provides a better handle of the accurate precision of the model. In relation to the number of TP and TN, 970 and 960, respectively, it can be concluded that the model has a potential to perform rather accurately when it comes to bird presence and absence detection. The comparatively low numbers for both FP and FN (30 and 40, respectively) also suggest the model is useful in reducing false positive results. The high predictive accuracy combined with the inference times described in this study situates the proposed model as a preferred candidate for real-time bird detection in safety-related aspects of aviation.

Figure 7 presents the training and validation accuracy over epochs, where both accuracies exhibit an upward trend, reflecting the model’s improving capability to correctly classify birds. The close tracking of validation accuracy with training accuracy implies that the model generalizes well to unseen data.

Figure 8 showcases the training and validation loss over epochs, depicting a steady decrease in loss values as the training progresses. This trend indicates that the model is learning effectively, with both training and validation losses converging, suggesting a good fit without overfitting the training data.

Figure 9a shows that the model is sensitive and precise in identifying birds as this can be determined using bounding frames which have been labeled to a single bird. Such customization of the detection parameters is indicative of the robustness of the feature extraction layers of the model which is significant for the accuracy of a program as in this case for real-time online applications. The accuracy of the model according to Table 1 is demonstrated with a precision of 97%, wherein this figure helps the reader to see the accuracy of the model with regards to the localizing of birds in a visual scene.

Figure 9b shows how the model can be used to predict multiple birds flying next to each other. This figure emphasizes another important strength of the model: the ability to forecast crowded scenes with all birds properly separated from one another—an important quality for operational systems considering the high birds per frame rates expected in scenes like those in an airport. The reliability of the accuracy estimated is 96.5% and the recall is 96%. These are highlighted in Table 1 and it shows the model’s ability to replicate the outputs with high detection under adverse conditions.

## 4. Discussion

This discussion is based on the applicability of the proposed spatiotemporal convolutional neural network (ST-CNN) which plays a significant role in the improvement in bird strike prevention technologies by detecting birds in a real-time manner. This advanced model is distinguished by the fact that it introduced both spatial data processing as well as temporal data processing, which is crucial to identify and track birds in dynamic airport environments.

The proposed model integrates an ATAN which targets the temporal aspect of the data with a SACN targeting spatial aspect. This design enables the detection to be precise by highlighting features of birds in backgrounds and their movements, a tremendous enhancement from the previous bird detection systems. This creates spatial attention mechanisms that allow the model to focus on concentrating on vital aspects of birds such as wing flapping or the changes in direction, thus improving the model’s ability to detect birds in environments that have different amounts of noise or interference.

In comparison to the current approaches, the benefit of this model can be supported by the results in terms of performance given in Table 1 of this paper. Specifically, it gains a huge advantage against the Faster R-CNN, Retina Net, and YOLO series models in terms of precision, recall, and IoU. This superior performance shows that the proposed ST-CNN is not only more accurate but also gains superior speed in processing and responding to real-time scenarios, and this is crucial in systems that are used in areas that require immediate action such as aviation safety systems.

In addition, deep learning to achieve the real-time interpretation of complex data in visual forms is proven to be considerable progress in the field. The function of decreasing false positives and increasing the detection accuracy presents a valuable tool to airports, which might diminish the high risk associated with bird strikes’ expenses while increasing the safety features. In this discussion, aware of the impacts of deep learning simulation on the risk management of the aviation industry by minimizing human errors of the pilots, the deep learning framework is proposed as a strong solution to the conventional detection systems in aviation which have been challenged by this safety issue for a long time.

Despite its robust performance, the proposed model has limitations primarily related to its generalization capabilities and real-time processing constraints. While effective in diverse scenarios, its accuracy may diminish in environments drastically different from those in the training dataset, particularly under extreme weather conditions or highly cluttered backgrounds. Additionally, the computational demands of the model, while manageable on high-end hardware, could pose challenges when scaling to broader, real-time applications requiring ultra-low latency. These areas present opportunities for further optimization and testing to enhance the model’s practical deployment in aviation safety and beyond.

## 5. Ablation Study

The ablation study aims to dissect the contributions of two critical components within the proposed model: ATAN and SACN. Table 4 presents the ablation study of the proposed methodology. The study evaluates the necessity and impact of each component by observing changes in performance metrics under three distinct configurations:iProposed Model: Integrates both ATAN and SACN, serving as the control setup to benchmark the full capabilities of our proposed system.iiWithout ATAN: Omits the ATAN component, leveraging only SACN to assess the importance of spatial feature analysis in the absence of temporal dynamics.iiiWithout SACN: Removes SACN, relying solely on ATAN to determine the impact of temporal analysis when spatial features are not considered.

Each model configuration is rigorously tested under uniform conditions using the same dataset, ensuring that any variations in performance are attributable solely to the changes in model architecture.

Proposed Model: The performance of the full model, incorporating both ATAN and SACN, sets a high standard across all evaluated metrics, including precision, recall, F1 score, accuracy, AUC-ROC, and IoU. This configuration highlights the synergistic effect of combining spatial and temporal analyses, crucial for the dynamic detection of objects such as birds in flight. The integration ensures robust detection capabilities, minimizing both false positives and false negatives, which is vital for applications requiring high reliability.

Without ATAN: This configuration shows a significant reduction in performance across all key metrics. The decrease in recall and precision is particularly notable, underscoring the critical role of temporal dynamics in tracking and identifying objects over time. The absence of ATAN suggests that temporal features are essential for understanding object behavior and movement, which are key to reducing identification errors and improving model responsiveness to real-time changes.

Without SACN: The removal of SACN results in the most dramatic decline in performance, especially in metrics like precision and IoU. This outcome emphasizes the importance of spatial analysis in accurately distinguishing objects from their backgrounds. Spatial features provide the necessary context for the model to localize and classify objects effectively within their environments. The pronounced drop in performance metrics indicates that spatial analysis is not merely supportive but fundamental to the model’s detection capabilities.

## 6. Conclusions

This research has successfully demonstrated the development and implementation of a novel deep learning framework for real-time bird detection, aimed at mitigating the risks associated with bird strikes in aviation. Through the integration of a spatiotemporal convolutional neural network, enhanced with spatial attention mechanisms and temporal analysis, the proposed model has shown superior performance in accurately identifying birds within diverse environmental settings. The experimental results provided a comprehensive evaluation of the model’s efficacy. The performance comparison with other meta-architectures which include Faster R-CNN, R-FCN, Retina Net, SSD, YOLO v4, and YOLO v5 shows that the proposed model achieved higher mean precision, mean recall, mean F1 score, mean accuracy, and maximum mean IoU. In addition, the inference time analysis supports the choice of the proposed model as a real-time solution, which is vital for the aviation safety systems. Such analysis is available in the confusion matrix; hence the revealed data show that the predictability of the model is very reliable in differentiating between birds and other objects. This capability is very important for the reduction in false alarms and activation of protective measures only when it is necessary. Possible further research can be devoted to the explanation of the addition of other data sources, such as radar and acoustic data for enhanced detection. Another area that could be further investigated included exploring the transfer learning and domain adaptation which could increase the model’s range of applications within the different geography and operation environments.

## Figures and Tables

**Figure 1 sensors-24-05455-f001:**
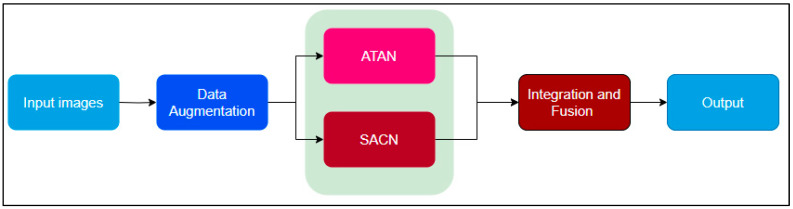
Proposed workflow.

**Figure 2 sensors-24-05455-f002:**
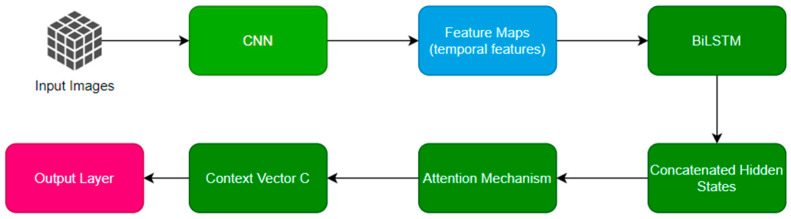
Workflow of ATAN.

**Figure 3 sensors-24-05455-f003:**
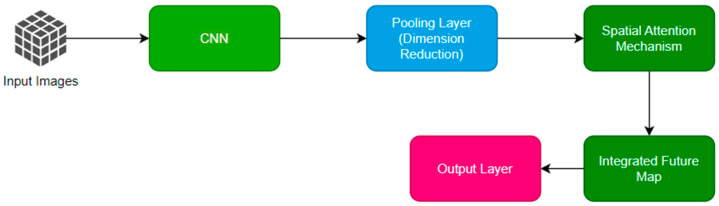
Workflow of SACN.

**Figure 4 sensors-24-05455-f004:**
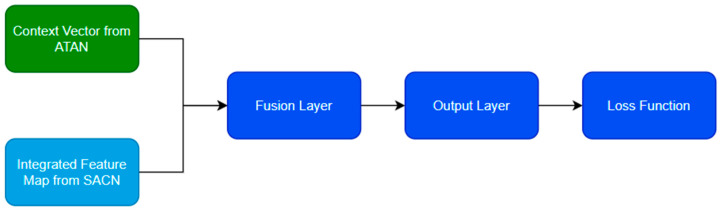
Workflow of integration and fusion.

**Figure 5 sensors-24-05455-f005:**
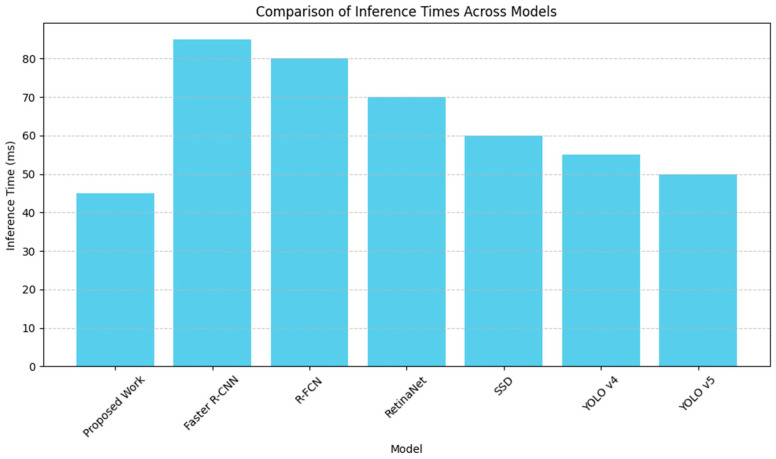
Comparison of inference times across models.

**Figure 6 sensors-24-05455-f006:**
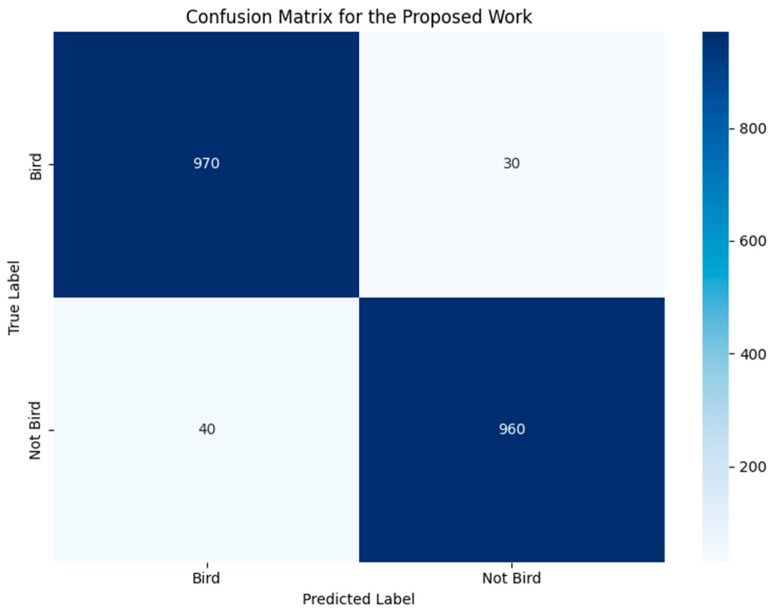
Confusion matrix of proposed model.

**Figure 7 sensors-24-05455-f007:**
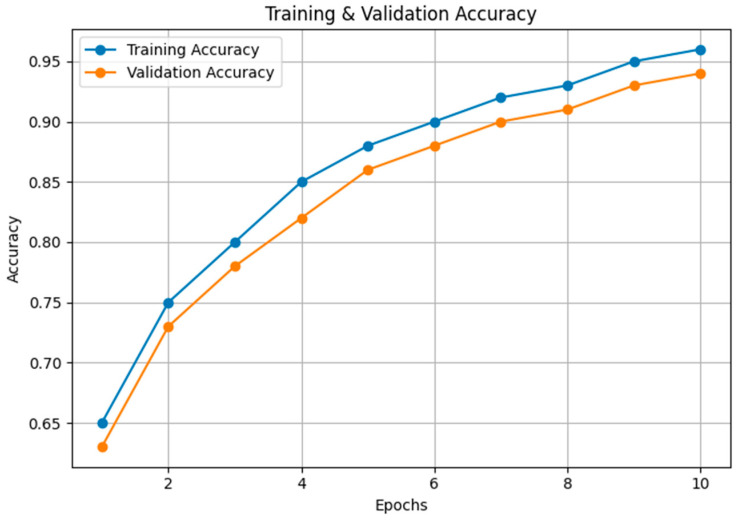
Comparison of training and validation accuracy.

**Figure 8 sensors-24-05455-f008:**
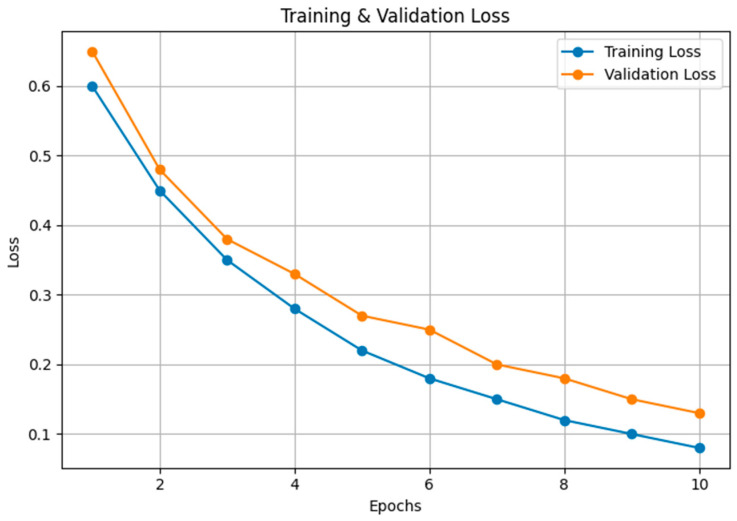
Comparison of training and validation loss.

**Figure 9 sensors-24-05455-f009:**
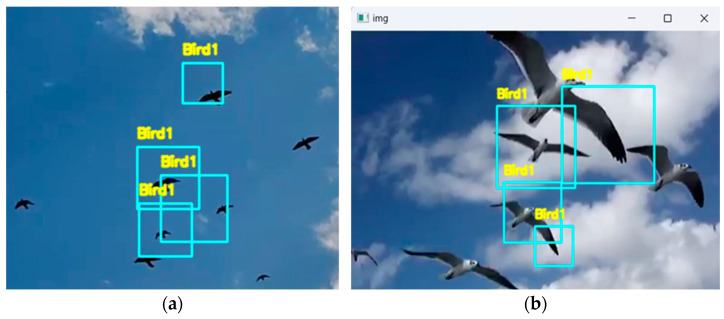
(**a**) Multiple detections for a single bird instance. (**b**) Effective bird detection in a dense scene.

**Table 1 sensors-24-05455-t001:** Data preprocessing parameters.

Parameter	Description	Example Values
Image Scaling	Rescaling images to a uniform size for model input.	256 × 256 pixels
Rotation	Rotating images to increase model robustness to orientation changes.	0°, 90°, 180°, 270°
Lighting Alteration	Adjusting brightness and contrast to simulate different lighting conditions.	Brightness ± 30%, Contrast ± 20%
Weather Simulation	Applying filters to mimic weather effects like fog, rain, or overcast conditions.	Fog intensity of 0.5, Rain overlay
Distance Variation	Zooming in or out to simulate different distances from the camera.	Scale images from 0.5× to 2×

**Table 2 sensors-24-05455-t002:** Hardware and software specifications.

Specification Type	Details
**Hardware**	
CPU	AMD Ryzen 9 7000 series
GPU	NVIDIA RTX 4070 Ti 16 GB
RAM	64 GB
Storage	2 TB NVME SSD
**Software**	
Operating System	Ubuntu 24 LTS
Deep Learning Framework	TensorFlow 2.×/PyTorch 1.× (specify as used)
**Deep Learning Parameters**	
Batch Size	32
Learning Rate	0.001 (adjust based on the optimizer)
Optimizer	Adam
Epochs	50
Loss Function	Cross-Entropy Loss (for classification tasks)

**Table 3 sensors-24-05455-t003:** Performance metrics’ comparison.

Model/Metric	Precision	Recall	F1 Score	Accuracy	AUC-ROC	IoU
Proposed Work	0.97	0.96	0.965	0.965	0.98	0.95
Faster R-CNN	0.90	0.89	0.895	0.92	0.95	0.80
R-FCN	0.88	0.87	0.875	0.91	0.93	0.78
Retina Net	0.89	0.88	0.885	0.90	0.94	0.79
SSD	0.87	0.86	0.865	0.89	0.92	0.77
YOLO v4	0.91	0.90	0.905	0.93	0.96	0.81
YOLO v5	0.92	0.91	0.915	0.94	0.97	0.82

**Table 4 sensors-24-05455-t004:** Ablation study.

Configuration	Precision	Recall	F1 Score	Accuracy	AUC-ROC	IoU
Proposed Work	0.97	0.96	0.965	0.965	0.98	0.95
Without ATAN	0.90	0.88	0.89	0.89	0.93	0.90
Without SACN	0.87	0.85	0.86	0.86	0.91	0.87

## Data Availability

The data supporting the findings of this study are available from the publicly archived dataset referenced as follows: Srivastav, Aditya; Shandilya, Shishir Kumar; Datta, Agni; Yemets, Kyrylo; Nagar, Atulya (2023), “Segmented Dataset Based on YOLOv7 for Drone vs. Bird Identification for Deep and Machine Learning Algorithms”, Mendeley Data, V3, available online: https://data.mendeley.com/datasets/6ghdz52pd7/5 (DOI: 10.17632/6ghdz52pd7.5, accessed on 5 December 2023). This dataset includes segmented images used for training and evaluating the deep learning models discussed in this paper. No new data were created during the study.

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
