# Peer review of "A Deep Learning Framework for Real-Time Bird Detection and Its Implications for Reducing Bird Strike Incidents"

_sensors, 2024, doi:10.3390/s24175455_

Round 1
Reviewer 1 Report
Comments and Suggestions for Authors
This paper have a research on "a Deep Learning Framework for Real-Time Bird Detection and Its Implications for Reducing Bird Strike Incidents“ which is a good issue. The paper builded a spatiotemporal convolutional neural network which employs spatial attention structures together with dynamic temporal processing to precisely recognize flying birds, which have some innovations. The deficiencies of the paper are as follows:
1. the context of line 139-147 should be deleted;
2. what's the DRL model presented in line 150? please give more description.
3. what's the output of the model ATAN and SACN? the probability distribution or context vectoe C and Fn. It's confusing in section 2.2.
4. please give supplementary experimental results comparison with the reference 22 and some ablation experiment for two model ATAN and SACN.
Author Response
This paper have a research on "a Deep Learning Framework for Real-Time Bird Detection and Its Implications for Reducing Bird Strike Incidents“ which is a good issue. The paper builded a spatiotemporal convolutional neural network which employs spatial attention structures together with dynamic temporal processing to precisely recognize flying birds, which have some innovations. The deficiencies of the paper are as follows:
- the context of line 139-147 should be deleted;
Thank you for notice. The context of line is deleted.
- what's the DRL model presented in line 150? please give more description.
Thank you for your inquiry regarding the mention of a "DRL model" in line 150 of our manuscript. Upon reviewing the text and the context in which this term was used, it appears to have been a typographical error. The correct terminology is "DL model," referring to "Deep Learning" models. Our paper employs deep learning methodologies involving convolutional neural networks and spatiotemporal processing techniques, specifically designed for the real-time detection of birds. These models leverage supervised learning mechanisms and do not incorporate reinforcement learning strategies. It was a typo mistake, once again thanking you for notice it.
- what's the output of the model ATAN and SACN? the probability distribution or context vectoe C and Fn. It's confusing in section 2.2.
Thank you for seeking clarification about the outputs of the ATAN (Attention-Based Temporal Analysis Network) and SACN (Spatially Aware Convolutional Network) models. Here is a detailed explanation to address the confusion:
- ATAN Output: The output of the ATAN model is a context vector C, which encapsulates the temporal dynamics of the observed bird movements. This context vector is computed using an attention mechanism that assigns weights to different temporal features based on their relevance for bird detection. The final output of ATAN is a dynamic representation of these features, which is then used to generate a probability distribution for the classification task—specifically, determining the presence or absence of birds.
- SACN Output: In contrast, the SACN model processes spatial features from individual frames and outputs feature maps . These feature maps are then enhanced through spatial attention mechanisms to better highlight bird-specific features against complex backgrounds. The final output from SACN, similar to ATAN, involves using these processed feature maps to provide a probability distribution over potential classifications in each frame.
Both outputs contribute to the model’s final decision-making process by integrating the highlighted temporal and spatial features, ensuring a robust and accurate detection system.
- please give supplementary experimental results comparison with reference 22 and some ablation experiments for two models ATAN and SACN.
Thank you for your suggestion to provide supplementary experimental results comparing our model with the systems developed using the dataset described in reference [22]. Upon reviewing the cited reference, we noticed that it is primarily a data article that introduces a segmented dataset for drone versus bird detection and does not present specific experimental results or performance metrics of detection models.
Reviewer 2 Report
Comments and Suggestions for Authors
1. Can you clarify what makes your model unique in the abstract?
2. Please discuss the limitations of previous models and how your model overcomes these issues in the background section.
3. Expand the literature review to include studies from the current year.
4. Provide detailed descriptions of the data preprocessing steps, including the specific algorithms used for image normalization and augmentation.
5. Include detailed hardware and software specifications in your experiments to help others reproduce your research.
6. Add a subsection that discusses potential errors or limitations in your model’s performance to provide insights into its reliability and areas for improvement.
7. Improve the clarity of the figures by adding comprehensive legends and labels to make them easier to understand.
8. Explain the comparison metrics like AUC-ROC and IoU in more detail and discuss why they are important for evaluating your model's performance.
9. Strengthen the conclusion by clearly summarizing the main findings and the impact of your research on the field.
Comments on the Quality of English LanguageIn the manuscript a few sentences need to rephrased to address the scientific rigor by removing ambiguity.
Author Response
- Can you clarify what makes your model unique in the abstract?
We revised the abstract to clearly highlight the unique aspects of the model, such as the integration of spatial and temporal analyses through ATAN and SACN, which differentiates it from existing models by enhancing real-time detection capabilities in complex environments.
- Please discuss the limitations of previous models and how your model overcomes these issues in the background section.
We expanded the background section to include a critique of previous models, specifically addressing their limitations in handling dynamic object detection in cluttered backgrounds. (line 97 – 102).
- Expand the literature review to include studies from the current year.
We included the recent papers in the background section of the paper.
- Provide detailed descriptions of the data prepossessing steps, including the specific algorithms used for image normalization and augmentation.
We elaborated on the data preprocessing steps in the methodology section, including detailed descriptions in the table - 1.
- Include detailed hardware and software specifications in your experiments to help others reproduce your research.
We provided detailed specifications of the hardware and software used in the experiments, including model training and testing environments in the table – 2.
- Add a subsection that discusses potential errors or limitations in your model’s performance to provide insights into its reliability and areas for improvement.
Limitations of the propsoed work is added under the discussion that critically evaluates potential errors and limitations in the model’s performance. (line 471 – 479).
- Improve the clarity of the figures by adding comprehensive legends and labels to make them easier to understand.
We enhanced the clarity of all figures by adding comprehensive legends and labels, ensuring they are self-explanatory.
- Explain the comparison metrics like AUC-ROC and IoU in more detail and discuss why they are important for evaluating your model's performance.
We provided a more detailed explanation of key comparison metrics such as AUC-ROC and IoU in the methodology section.
- Strengthen the conclusion by clearly summarizing the main findings and the impact of your research on the field.
We revised the conclusion to more clearly summarize the main findings and emphasize the impact of the research on the field. This will involve discussing the practical implications of the model and its potential to advance current practices in object detection.
Round 2
Reviewer 1 Report
Comments and Suggestions for Authors
In the version-v2,some figures are not clear as v1, for example figure 1,figure 2 and figure 5, they are too hazy.
Comments on the Quality of English LanguageThe English in the manuscript can correctly express the author's meaning.
It might be better if it could be touched up.
Spatiotemporal Convolutional Neural Network (ST-CNN) ->
Spatio-Temporal Convolutional Neural Network (ST-CNN)
Reviewer 2 Report
Comments and Suggestions for Authors
Nil